# Inter and transgenerational impact of H3K4 methylation in neuronal homeostasis

Steffen Abay-Nørgaard[1] , Marta Cecylia Tapia[1,2], Mandoh Zeijdner[1] , Jeonghwan Henry Kim[1] , Kyoung Jae Won[1], Bo Porse[1,2,3], Anna Elisabetta Salcini[1]

**Epigenetic marks and associated traits can be transmitted for one or more generations, phenomena known respectively as inter- or transgenerational epigenetic inheritance. It remains unknown if genetically and conditionally induced aberrant epigenetic states can influence the development of the nervous system across generations. Here, we show, using *Caenorhabditis elegans* as a model system, that alteration of H3K4me3 levels in the parental generation, caused by genetic manipulation or changes in parental conditions, has, respectively, trans- and intergenerational effects on H3K4 methylome, transcriptome, and nervous system development. Thus, our study reveals the relevance of H3K4me3 transmission and maintenance in preventing long-lasting deleterious effects in nervous system homeostasis.**

## Introduction

Lying at the interface between the genome and the environment, the epigenome modulates transcriptional programs during development and throughout the life of an organism, in response to endogenous and exogenous cues. Epigenetic changes, including histones/DNA modifications and small RNAs, can alter genome functions without changing the underlying DNA sequence resulting in epigenetic phenotypes. Epigenetic phenotypes can span from one to numerous generations and their persistence over generations is used to define the epigenetic inheritance as intergenerational (if the epigenetic phenotype lasts one/two generations and derived from germ cell directly exposed to the cues) and transgenerational (if the epigenetic phenotype lasts beyond three generations and it is decoupled from germ cell exposure to the cues) (Perez & Lehner, 2019). How and if epigenetic modifications, and the memory of experiences they might encode, are transmitted together with epigenetic traits across generations are key questions

in biology that recently have attracted much attention (Liberman et al, 2019; Perez & Lehner, 2019; Senaldi & Smith-Raska, 2020; Fitz-James & Cavalli, 2022).

In the context of neuronal development and neuronal diseases, epidemiological studies suggest that parental conditions could have an impact on the formation of the nervous system in children (Parner et al, 2012; Noble et al, 2015; Sandin et al, 2016; Golding et al, 2017; Kim et al, 2021; Tooley et al, 2021) and that grandparental age or tobacco consumption might be correlated to neurodevelopmental disorders in grandchildren, such as autism (Frans et al, 2013; Golding et al, 2017; Gao et al, 2020; Xiao et al, 2021). Whether environmental stimuli increase the risk of neurodevelopmental disorders is currently under debate (Ciptasari & van Bokhoven, 2020). On the other hand, the establishment and maintenance of a proper epigenetic landscape is considered of paramount importance during development of the nervous system, as inherited or de novo mutations in chromatin regulators represent the second most-associated category found in neurodevelopmental disorders (LaSalle et al, 2013; De Rubeis et al, 2014; Pinto et al, 2014; Iwase et al, 2017; Gabriele et al, 2018; Satterstrom et al, 2020; Mossink et al, 2021; Reichard & Zimmer-Bensch, 2021). Among them, mutations in almost all members of the COMPASS complex, a highly conserved multisubunit complex that methylates histone 3 Lysine 4 (H3K4), are frequently reported (Miller et al, 2001; Robert et al, 2014; Collins et al, 2019). The importance of the trimethyl form of H3K4 (H3K4me3), associated with active gene transcription (Liu et al, 2005; Barski et al, 2007; Zhang et al, 2009; Beurton et al, 2019), for proper brain development is also testified by significant changes during cerebral cortex remodeling in early human life (Shulha et al, 2013) and by studies in mouse models in which deregulation of H3K4 methylation patterns results in neuronal developmental defects and memory deficits (Kim et al, 2007; Gupta et al, 2010; Kerimoglu et al, 2013; Ancelin et al, 2016; Wasson et al, 2016).

With conserved epigenetic machinery and a short generation time, *Caenorhabditis elegans* (*C. elegans*) has proven to be an invaluable tool to study epigenetic inheritance (Perez & Lehner,

[1]Biotech Research and Innovation Centre, Faculty of Health Sciences, University of Copenhagen, Copenhagen, Denmark   [2]The Finsen Laboratory, Rigshospitalet, University of Copenhagen, Copenhagen, Denmark   [3]Novo Nordisk Foundation Center for Stem Cell Biology, DanStem, Faculty of Health Sciences, University of Copenhagen, Copenhagen, Denmark

Correspondence: lisa.salcini@bric.ku.dk; noerrebro@hotmail.com
Jeonghwan Henry Kim's present address is School of System Biomedical Science, Soongsil University, Seoul, Korea
Kyoung Jae Won's present address is Department of Computational Biomedicine, Cedars-Sinai Medical Center, West Hollywood, CA, USA

2019; Baugh & Day, 2020). A body of recent literature revealed how different behaviors, such as chemotactic and pathogenic avoidance, can be inherited in a transgenerational manner via small RNA-related mechanisms (Moore et al, 2019; Posner et al, 2019). As all behaviors are rooted in the nervous system, it is reasonable to postulate that epigenetic changes might control neurodevelopment across generations. H3K4 methylation, mainly deposited by the catalytic subunits of the COMPASS complexes SET-2 (SETD1A-B/KMT2E-F) and SET-16 (MLL1-4/KMT2A-D) (Pedersen & Helin, 2010, Xiao et al, 2011), and removed by RBR-2 and SPR-5 demethylases (Christensen et al, 2007; Katz et al, 2009), has been suggested to function as a molecular memory in *C. elegans* because perturbation of H3K4 level has transgenerational effects on fertility, lifespan, and fat storage (Katz et al, 2009; Greer et al, 2011, 2016; Nottke et al, 2011; Han et al, 2017; Wan et al, 2022) but, so far, not implicated in the transmission of neurodevelopmental defects over generations.

We previously reported that the H3K4me3 regulatory machinery controls nervous system development in *C. elegans* (Mariani et al, 2016; Riveiro et al, 2017; Abay-Nørgaard et al, 2020) by showing that proper regulation of H3K4me3 levels is required for correct axon extension of PVQs, two bilateral interneurons that project their axons in the left and right fascicles of the ventral nerve cord during embryogenesis (Durbin, 1987; Bülow et al, 2004; Boulin et al, 2006; Torpe & Pocock, 2014). Here, we show that the developmental PVQ defects derived by genetic perturbation of H3K4me3 levels are transgenerationally inherited, together with aberrant H3K4me3 levels and transcriptome, establishing a model for understanding how heritable chromatin modifications affect neurodevelopment. We also provide evidence that parental conditions can embed H3K4me3 changes in the epigenome, with intergenerational effects on PVQ axon guidance and transcription that are SET-2 dependent. Thus, perturbation of H3K4 methylation level, related to mutations in H3K4 regulators and to certain conditions, can have inter and transgenerational consequences on neuronal development.

## Results

### H3K4 methylation-related axon guidance defects can be inherited in a transgenerational manner

We previously demonstrated that loss/inactivation of most of the H3K4 methylation regulators, including *set-2*, *set-16*, and core components of the COMPASS complexes, resulted in failure of the PVQ neurons to properly project their axons (here defined as PVQ axon guidance defect/phenotype, Fig 1A) (Abay-Nørgaard et al, 2020). To study the inheritance pattern of the *set-2*-related PVQ phenotype across generations, we used a genetic crossing strategy (Fig 1B). We crossed WT males with *set-2(zr1208)* hermaphrodites (carrying a point mutation in a conserved residue of the SET domain [Y1397F] resulting in strong H3K4me3 decrease [Fig S1A]) and examined the PVQs in cross-generated descendant lines, either homozygous mutant (MUT-des) or wild type (WT-des). As homozygous lines are established in the second generation (F2), we analyzed the third generation (F3) (Fig 1B). We observed remarkable PVQ phenotypic heterogeneity in MUT-des lines after outcrossing

(Fig 1C, see the Materials and Methods section for statistical approach). However, the lines gradually acquired the PVQ defects over generations and all presented a significant PVQ phenotype at the F6 generation (Fig 1C). From the same outcross, we followed independent WT-des lines. Seven out of the 40 WT-des F3 lines tested (17.5%) exhibited significant PVQ defects (Fig 1D). One of the WT-des lines had a persistent PVQ phenotype up to F4 (Fig S1B) that disappeared at F5 (8%, n = 50). The evidence that, at F5, all WT-des lines show no PVQ defects indicates that these defects are not linked to an extraneous mutation. The persistent PVQ phenotype was also observed in F3 WT-des when *set-2(zr1208)* mutant males were crossed with WT hermaphrodites (Fig 1C), although at low penetrance, and when using another catalytic-inactive *set-2* mutant allele for the crossing, *set-2(zr2012)* (Figs 1E and S1A and D). Overall, these results suggest that a parental mutation in the catalytic domain of *set-2* can affect PVQ axon guidance for several generations, despite the descendants carrying WT alleles. To reinforce the notion that this effect is related to changes on H3K4 methylation, we used the same cross strategy with an *ash-2* mutant, a core component of the COMPASS complex, also reported to have a PVQ defect (Abay-Nørgaard et al, 2020) (mean ± sem: 35% ± 1.7%, n = 150) and to affect H3K4me3 levels (Miller et al, 2001). WT-des lines derived from *ash-2* crosses show inheritance of the PVQ defects in F3 and F4 (Figs 1F and S1E). In one line, the phenotype persisted in F5 and disappeared in F6 (F5 18%, n = 50, F6 6%, n = 50). To exclude the possibility that the phenotypic heterogeneity observed could be related to the crossing procedure or to statistical limitations, we crossed WT males with WT hermaphrodites and analysed the PVQs in descendant lines (wt-ctrl). All 38 wt-ctrl lines tested did not show statistically significant PVQ defect at F3 generation (Fig 1G). Finally, we analysed F3 WT-des lines derived from crossing WT males with *unc-6* mutant hermaphrodites, lacking netrin, a signaling molecule required for proper PVQ axon guidance, with a highly penetrant PVQ phenotype (mean ± sem: 82% ± 1.8%, n = 150) (Hutter, 2003). All 26 F3 WT-des lines analysed showed no PVQ defects (Fig 1H), indicating that the persistence of the PVQ phenotype over generations is specific for mutants of H3K4me regulators.

These results indicate that the PVQ phenotype, observed when H3K4 methylation machinery is compromised, can be transgenerational inherited in genotypically WT descendants.

### Aberrant H3K4me3 levels are transmitted and affect the transcriptome over generations

Our crossing experiments suggest the presence of a *set-2*-dependent information passing between generations, required for proper PVQ development. Given that small RNAs have previously been linked in the inheritance of environmentally induced phenotypes in *C. elegans* (Rechavi et al, 2014; Posner et al, 2019; Toker et al, 2022), we tested if the mediator of small RNA inheritance, *hrde-1*, was implicated in the inheritance of the PVQ phenotype. Mutants of *hrde-1* (deficient in inheriting small RNAs (Rechavi et al, 2014) do not show PVQ axon defects (Fig S1F). Furthermore, by using the same crossing strategy described above with a *set-2 hrde-1* double mutant, we observed persistent PVQ defects in F3 WT-des lines, with eight out of 40 lines (20%) inheriting the PVQ defects (Fig 1I). Therefore, we concluded that that small RNAs are not

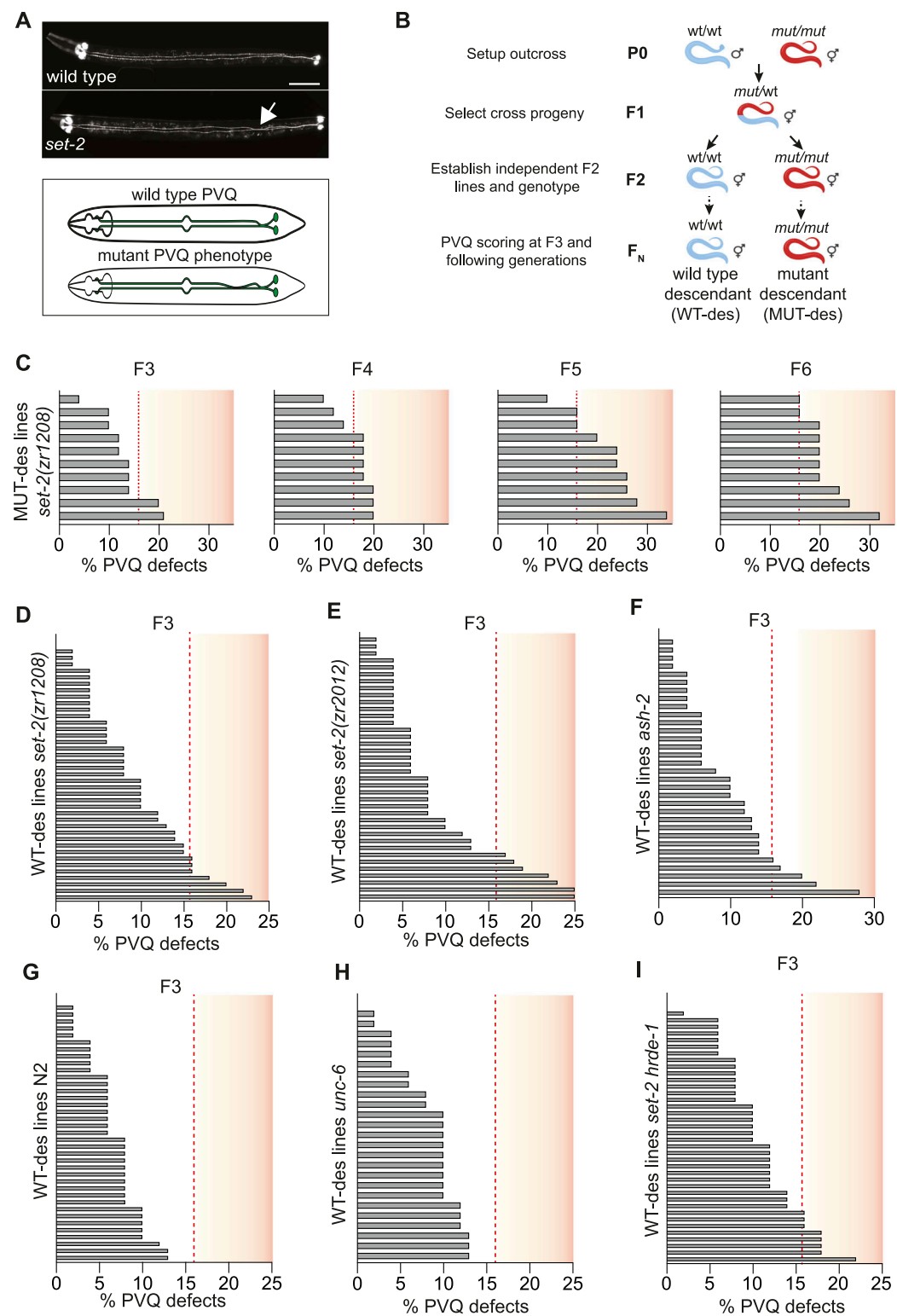

**Figure 1. Transgenerational effects of H3K4me3 modifiers on PVQ development.**

**(A)** Top: ventral view of transgenic animals expressing *oyIs14* fluorescent reporter visualizing PVQ neurons in WT (top) and in *set-2(zr1208)* mutant (bottom). In WT, PVQ cell bodies are positioned posteriorly and their axons project anteriorly on each side of the ventral midline. In *set-2* mutants, the axon of left PVQ fails to respect the midline and defasciculates to the right ventral nerve cord (designated by white arrow) to later returns. Posterior is on the right. Scalebar 50 $\mu$m. Bottom: schematic depiction of WT and mutant PVQs. **(B)** Schematic depiction of the outcrossing strategy employed to follow wild-type descendants (WT-des) and mutant descendants. $P_0$ is the crossed parental generation. $F_1$ is the first generation where cross-progeny is selected, and hermaphrodites are left to self-fertilize. $F_2$ is the second generation from which independent lines are established. $F_N$ is the third and subsequent generations. **(C)** PVQ defects at indicated generations of independent mutant descendants lines

required for the inheritance of the PVQ phenotype observed in WT descendants. Complete data of the crossing experiments are presented in Table S1 and statistical analyses at total population level are summarized in Fig S1G. It should be noted that, at the population level, *set-2*, *ash-2,* and *set-2 hrde-1* mutants show significant PVQ defects compared with wt-ctrl and *unc-6,* but that the inheritance through male is not statistically significant, suggesting that the inheritance is likely occurring through the oocyte.

We next asked if H3K4me3 is the transmitted signal and whether a parental *set-2* mutation has effects on the H3K4me3 levels across generations. To this end, we examined the global level of H3K4me3 in WT-des and MUT-des lines derived from crossed *set-2(zr1208)* at F3 and F5 generations by spike-in H3K4me3 chromatin immuno-precipitation and sequencing (ChIP-seq). We failed to obtain enough material from mid-embryos, when PVQ axons extend, therefore, we used chromatin extracted from L4 stage animals yielding H3K4me3 profiles comparable with those previously reported Jänes et al (2018). We identified a set of high-confidence H3K4me3 peaks in each condition and combined these to generate a union of peaks (all peaks) which were used for downstream analysis. As expected, principal component analysis (PCA) of ChIP-seq data revealed that WT-des and MUT-des were in distinct clusters (Fig 2A), and that H3K4me3 levels were reduced in mutant descendants compared with WT descendants at any generation tested (Fig 2B and C). Heatmaps and tracks of H3K4me3 in different WT-des and MUT-des lines are shown in Figs S2A and B.

A minimal increase of H3K4me3 levels was observed in *set-2* mutant descendants at F5 generation compared with F3 (Fig 2B and C), suggesting a negligible compensatory contribution of other H3K4 methyltransferases to H3K4 deposition over generations. Surprisingly, in WT descendant lines, H3K4me3 levels increased substantially from F3 to F5 generations (Fig 2B and C) and more H3K4me3 peaks were recovered in F5, compared with F3 (Fig 2D), suggesting that H3K4me3, when lost, is slowly reestablished. The increase in H3K4me3 levels across generations of WT-des could also be seen when the analysis was limited to peaks common to both F3 and F5 WT-des (98% of all F3 peaks), indicating that the observed increase of H3K4me3 levels in the F5 WT-des is not only because of an increased number of H3K4me3 peaks in the F5 WT-des (Fig S2C and D). Furthermore, we found that the F5 WT-des H3K4me3 ChIPseq signal correlated stronger with previously published H3K4me3 ChIPseq data from WT L4 animals compared with F3 WT-des (Fig S2E and F). Taken together these results indicate that in WT descendants derived from an ancestral animal lacking SET-2 catalytic activity, the H3K4me3 signal increases over generations towards a more WT level.

To test if the different levels of transmitted H3K4 methylation signal affect the expression of protein coding genes, we performed RNA-seq in independent lines of WT (WT-ctrl, generated after crossing WT male with WT hermaphrodites), WT-des, and MUT-des, at F3 and F5 at the L4 stage. PCA of RNA-seq data revealed that the gene expression profiles of the F3 and F5 WT-des cluster in two similar, but distinct, transcriptomic groups (Fig 2E), with the F5 profile closer to WT-ctrl. Consistently, heatmaps of the expression levels of differentially expressed genes between WT-ctrl, WT-des, and MUT-des (Fig 2F) position F5 WT-des lines closer to WT-ctrl, suggesting that the transcriptomic profile of WT-des gradually, over generations, moves toward the pattern of expression of WT-ctrl. As we used L4 animals for these analyses, we did not identified genes involved in axon guidance, occurring during a specific stage of embryogenesis. Nevertheless, the results strongly suggest that WT descendants derived from a parental *set-2* mutant exhibit, at early generations, aberrant H3K4me3 level coupled to transcriptional defects.

### Parental conditions have intergenerational effects on axon guidance

Having established that proper PVQ axon guidance hinges on inherited H3K4me3 levels, we wondered if environmental conditions known to affect the epigenome, like food abundance and temperature (Rechavi et al, 2014; Ni et al, 2016; Klosin et al, 2017; Costello & Petrella, 2019; Delaney et al, 2019; Toker et al, 2022; Wan et al, 2022), could have a similar impact. We therefore examined the PVQ axon guidance in animals and/or their progeny exposed to starvation and high temperatures (Fig 3A). Whereas parental starvation had no effect on the PVQ neurons in F1 progeny (Fig 3B), temperature increase had a robust and immediate influence on PVQ development. When WT hermaphrodites at the L4 stage were shifted from the standard temperature of 20°C to a higher, still permissive, temperature of 25°C, defects in PVQ axon guidance were observed in their progeny (F1). Interestingly, the penetrance of the phenotype increased over time, reaching a plateau after two generations (Fig 3C). The PVQ phenotype was, however, not inherited in the progeny of animals resettled at 20°C (at L4 stage), after being exposed at high temperature for three generations (Fig 3C), indicating the need of a continuous environmental stimulus. To test the specificity of the PVQ defects, we analyzed the development of other neurons under the same conditions (Table S2). We observed axon guidance defects in PVP and HSN neurons at high temperature, whereas starvation only affected the migration of the HSN neurons cell body. Other classes of neurons were not disturbed in these conditions, indicating that the overall structure of

from WT males crossed with *set-2(zr1208)* hermaphrodites. The number of lines with defects increases over generations, from two lines in the third generation (F3) to all 10 lines in the sixth generation (F6). **(D)** PVQ defects in F3 WT-des independent lines, from WT males crossed with *set-2(zr1208)* hermaphrodites. **(E)** PVQ defects in F3 WT-des independent lines, from WT males crossed with *set-2(2012)* hermaphrodites. **(F)** PVQ defects in F3 WT-des independent lines, from WT males crossed with *ash-2(tm1905)* hermaphrodites. **(G)** PVQ defects in F3 WT-ctrl independent lines, from WT males crossed with WT hermaphrodites. Zero lines are significantly different compared with WT. **(H)** PVQ defects in F3 WT-des independent lines coming from WT males crossed with *unc-6(ev400)* hermaphrodites. Zero lines are significantly different compared with WT. **(I)** PVQ defects in F3 WT-des independent lines, from WT males crossed with *set-2(zr1208) hrde-1(tm1200)* hermaphrodites. In C–I, each grey bar represents the scoring from a single independent line coming from 1-d-old hermaphrodites (n = 50–52). Phenotypic threshold (red stippled line) is defined as the lowest PVQ penetrance required for an independent line (n = 50–52) to be significant different (*P* < 0.05) from the WT PVQ penetrance defect (6%, n = 200) using chi-square method or Fisher's exact test.

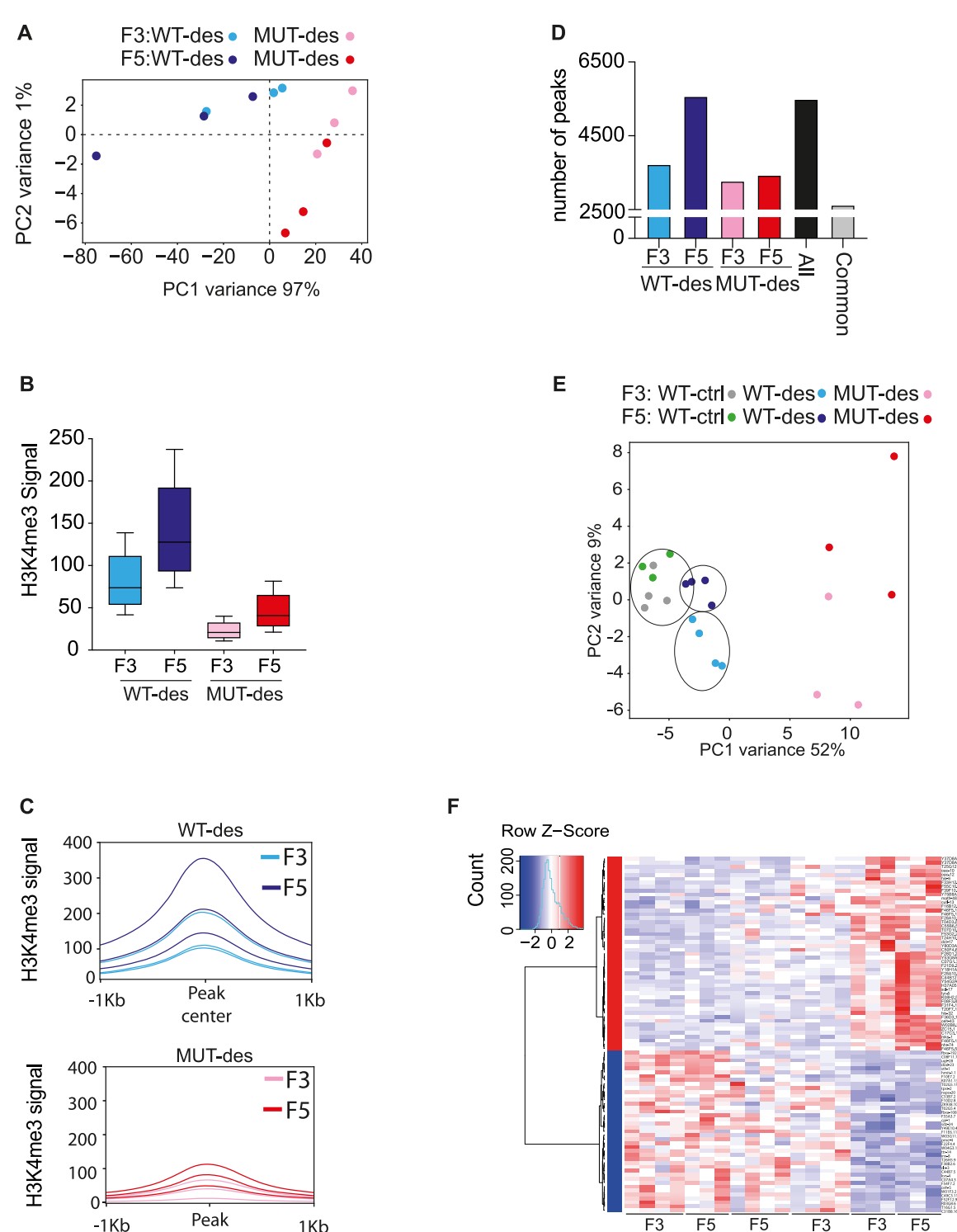

**Figure 2. H3K4me3 and transcriptome changes in descendants.**
Wild-type descendant (WT-des) and mutant descendant (MUT-des) lines originates from crossing WT males with *set-2(zr1208)* hermaphrodites. **(A)** Principal component analysis of H3K4me3 ChIP-seq data from WT-des and MUT-des lines at F3 and F5 generations. **(B)** Box plot of median H3K4 signal (all peaks) based on H3K4me3 ChIP-seq data from F3 and F5 generations in WT-des and MUT-des lines. Boxes are 25th to 75th percentile, whiskers represent min and max. **(C)** H3K4 profile plots based on H3K4me3 ChIP-seq data from independent samples from WT-des and MUT-des lines at F3 and F5 generations. **(D)** Number of H3K4me3 peaks in WT-des and MUT-des lines at F3 and F5 generation. All peaks: union of peaks found in all conditions. Common peaks: overlapping peaks in all conditions. **(E)** Principal component analysis of mRNA expression the L4 stage in WT-des, MUT-des, and WT-ctrl at F3 and F5 generations. Every dot is an independent line. WT-ctrl originates from WT males crossed with WT

the nervous system is not compromised by high temperature. We conclude that increased temperature affects axon guidance of a subset of neurons. However, the defect is not transmitted to the next generation when the external stimulus is removed.

We also tested the effect of parental age (Fig 3A), a physiological condition known to influence phenotypic traits in *C. elegans* (Perez et al, 2017) and associated with increased risk of neuro-developmental defects in humans (Parner et al, 2012; Merikangas et al, 2017). When the progeny of 3- and 4-d-old mothers was analyzed and compared with the progeny of 1-d old mothers, we observed axon guidance defects in the PVQ neurons in the F1 but not F2 (Figs 3D and S3A) progeny, with penetrance rising with increasing maternal age. Consistently, we found that genetic alteration of age (Kenyon et al, 1993) compromises axon guidance, and progeny from short-lived *daf-16* mutants showed a ~20% increase in axon guidance defects starting from 3-d-old adults (Fig S3B) and progeny of 4-d-old long-lived *daf-2* mutants did not show any significant PVQ defects associated with parental aging. Of note, *daf-16* mutant animals showed a high penetrant PVQ guidance phenotype also under normal condition. Advanced maternal age also compromised PVP axon guidance but no other neurons, indicating that PVQ and PVP neurons are particularly sensitive to this condition (Table S1). Overall, these data indicate that parental conditions, such as maternal age and high temperature, can have intergenerational effects in the axon guidance process of a subset of neurons.

### Conditionally induced PVQ axon phenotype correlates with increased H3K4me3 levels and aberrant transcription

As the penetrance and pattern of axon guidance defects observed at high temperature or in progeny of old mothers were strikingly similar to the ones found in *set-2* mutants (Abay-Nørgaard et al, 2020), we investigated if H3K4me3 levels change under these conditions. Unexpectedly, we found the global levels of H3K4me3 were slightly, though consistently, elevated in animals exposed to high temperature for two generations and, more evidently, in aged animals, in agreement with previous studies (Pu et al, 2018), in aged animals (Fig 3E). Interestingly, we also observed an H3K4me3 increase also in the offspring of old mothers (Fig 3E). Although it is possible that the increased level of H3K4me3 in F1 L4 is a consequence of defects related to aged oocytes (Perez et al, 2017), it is also tempting to assume that the F1 inherited a higher level of H3K4me3 from the mothers. Apart from an increase in H3K36me3, a posttranslational modification decorating, like H3K4me3, actively transcribed genes (Kizer et al, 2005), no other posttranslational modifications were changed in aged hermaphrodites and their progeny (Fig S3C and D), suggesting that the H3K4me3 increase observed in this condition is rather specific.

To test if these changes resulted in alterations of the transcriptome, we performed RNA-seq in F1 progeny (at the L4 stage) derived from 4-d-old mothers, from animals kept for one generation at high temperatures and in the F1 progeny generated after

the mothers (at the L4 stage) were returned to 20°C, with no PVQ defects (Fig 3C). PCA revealed that the transcriptome of the F1 progeny of animals exposed to high temperatures and aged animals was distinct from the progeny of 1-d-old mothers grown at normal temperature, even if previously exposed to high temperature (Fig 3F). In line with the increased H3K4me3 levels in animals exposed to high temperature and in progeny from old mothers, differentially expressed genes in these conditions showed a strong bias, with 83% and 99% of differentially expressed genes being up-regulated, respectively (Fig 3G). Taken together, these results show that certain parental conditions modify the H3K4me3 level and the transcriptional landscape, with an impact on neuronal development in the following generation.

### Conditionally induced PVQ axon phenotype is regulated by SET-2

To further investigate the involvement of H3K4 methylation and a possible role of *set-2* in the conditionally induced PVQ phenotypes, we tested the effects of aging and temperature increase in SET-2 catalytic inactive mutant animals. We also included in these experiments a catalytic inactive mutant of SET-16, the other known H3K4 methyltransferase in *C. elegans*, that shows PVQ defects under normal conditions (Fig 4A). With advanced maternal age, we detected a further increase in the penetrance of the PVQ guidance phenotype in *set-16* but not in the *set-2* mutant animals (Fig 4A). Similarly, when the two mutants were grown at 25°C for three generations, we detected an increase in the penetrance of the phenotype in *set-16* but not in the *set-2* mutant animals (Fig 4B), suggesting that *set-2* is responsible for the conditionally induced PVQ phenotype. In agreement with these results, we found that most of the up-regulated genes detected under these conditions (Fig 3G) are *set-2* dependent, as their expression returns to WT levels in progeny from old mothers (Fig 4C, left) or in the progeny of animals exposed to high temperatures (Fig 4C, right) with a *set-2* background.

As elevated H3K4me3 signal could be inherited in F1 descendants from aged mothers (Figs 3E and S3), we tested if aberrant H3K4me3 levels were contributed by the oocyte and/or the sperm, by crossing young WT males with old WT hermaphrodites and vice versa and analysed the PVQ defects in F1 cross-progeny. Cross-progeny from both setups showed normal PVQ development, whereas the cross-progeny of old WT male with old WT hermaphrodites resulted in a PVQ phenotype like the one observed in the self-progeny of aged hermaphrodites (Fig 4D). As animals used in these crosses are genetically WT, this result implies that the epigenetic contribution from one "young" gamete, either oocyte or sperm, is sufficient to restore proper PVQ development. In agreement, we found that all deregulated genes in F1 progeny derived by old mothers can be rescued by supplying young sperms (Fig 4C, right). Importantly, when young *set-2* males were crossed with old WT hermaphrodites, the penetrance of PVQ defects in the F1 cross progeny was like the one observed in old mothers' self-progeny, whereas crossing with *set-16* males completely rescued

hermaphrodites. **(F)** Heatmap based on RNA-seq data showing up- and down-regulated genes at the L4 stage in F3 and F5 descendants. The genes shown are differentially regulated between WT-ctrl (F3 and F5) and MUT-des (F3 and F5). Gene expression is shown as a relative Z-score across samples.

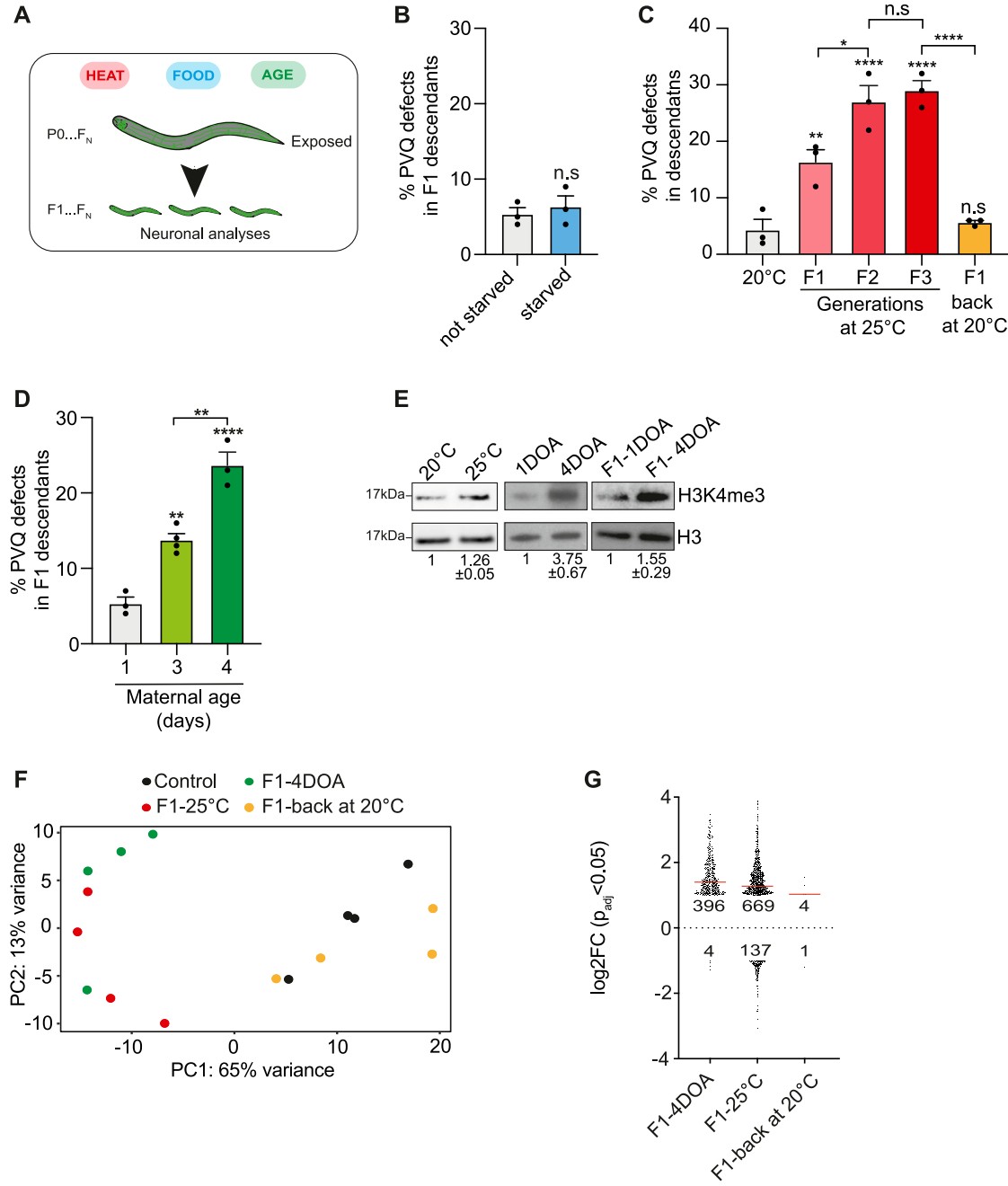

**Figure 3. Parental conditions can affect neuronal developmental, H3K4me3 level, and transcriptome in F1 descendants.**
**(A)** Schematic depiction of different conditions used. **(B)** PVQ defects in starved animals. WT animals were starved for 24 h at the L4 stage. n ≥ 150 pr. condition. n.s. not statistically significant comparing with progeny from non-starved animals. **(C)** PVQ defects in WT L4 animals exposed to heat (25°C) at indicate generations, and after moving the animals (F3 generation) back to low temperature (20°C). n ≥ 210 pr. condition. *P < 0.05, **P < 0.01, ****P < 0.0001, and n.s. not statistically significant, when comparing with WT grown at 20°C or between columns, as indicated by black lines. **(D)** PVQ defects in progeny in ageing WT animals (1, 3- and 4-d old adults). n ≥ 190 pr. condition. **P < 0.01, ****P < 0.0001, comparing with progeny from 1-d old hermaphrodites or between columns, as indicated by black lines. In (B, C, D), data are expressed as mean ± SEM. Each black dot represents an independent experiment. **(B, C, D)** Statistical significance was assessed using t test (B) or one-way ANOVA Tukey's multiple comparison (C, D). **(E)** Representative Western blot of H3K4me3 levels (L4 stage). Left: animals grown at 20°C and shifted at 25°C for two generations. Middle: 1-day-old adults and 4-day-old adults (4DOA). Right: progeny (F1) from 1DOA and 4DOA. H3 is used as loading control. Western blots were carried out at least three times. Numbers represents mean levels ± SEM. **(F)** Principal component analysis of mRNA expression under different conditions. Each dot represents an independent biological replicate in the RNA-seq analysis. Control (black) represents the progeny from 1-d-old animals grown at 20°C. Maternal age (green) represents progeny from 4DOA at 20°C. F1-25°C (red) represents the progeny from 1DOA exposed to 25°C for one generation. F1-back (yellow) represents the progeny from 1DOA exposed to 25°C for two generations and moved back to 20°C for one generation. **(G)** Differentially expressed (DE) genes in animals (L4 stage) exposed to different conditions. F1-4DOA, DE genes identified comparing the L4 progeny from 1DOA and from 4DOA. F1-25°C, DE genes identified comparing L4 progeny from animals kept at 20°C and 25°C for one generation. F1-back, DE genes identified comparing L4 progeny from animals kept at 20°C with animals kept at 20°C for one generation after being exposed to 25°C for two generations. Dots represents DE genes (log$_2$FC > ± 1, Padj < 0.05). Red bar is the average log$_2$FC. Numbers represent numbers of genes up- and down-regulated under indicated conditions.

**Figure 4. Alterations related to parental conditions are SET-2 dependent.**

**(A)** PVQ defects in the progeny from 1-day-old adults (1DOA) and from 4-day-old adults (4DOA) WT, *set-2(zr1208)* and *set-16(zr1804)*. n ≥ 150 pr. condition. *P < 0.05, ***P < 0.001, n.s. not statistically significant, comparing with the progeny from 1DOA same genotype or between columns, as indicated by black lines. **(B)** PVQ defects in WT, *set-2(zr1208)* and *set-16(zr1804)* animals grown at 20°C and at 25°C for three generations. n ≥ 150 pr. condition. ***P < 0.001, ****P < 0.0001, n.s. not statistically significant comparing 25°C with 20°C same phenotype or between columns, as indicated by black lines. **(C)** Left. Differentially expressed genes (log₂FC > ± 1, Padj > 0.05) based on RNA-seq, identified comparing the L4 progeny from 1DOA and 4DOA WT animals. log₂FC of the same genes in the L4 progeny from 1DOA and 4DOA *set-2(zr1208)* mutants (*set-2*) and in the L4 progeny from 4DOA hermaphrodites crossed with young males compared with progeny from 1DOA hermaphrodites crossed with young males (wt crossed). Right: differentially regulated genes (log₂FC > ± 1, Padj > 0.05) based on RNA-seq, identified comparing the L4 progeny from WT grown at 20°C and WT kept for one generation at 25°C (WT). log₂FC of the same genes in progeny of *set-2(zr1208)* mutants grown at 20°C and *set-2(zr1208)* kept one generation at 25°C (*set-2*). **(D)** PVQ

the PVQ phenotype (Fig 4E), suggesting that the genetic or epigenetic contribution derived by a "young" gamete carrying a *set-2* catalytic inactive allele, is not sufficient to restore proper PVQ development. Similar results were obtained analyzing PVQ defects derived when exposing to high temperature. The heat-induced PVQ defects were completely rescued in cross-progeny coming from hermaphrodites kept at 25°C for two generations crossed with young WT or *set-16* males grown at 20°C (Fig 4F). However, the genetic or epigenetic contribution derived by a "cold" gamete carrying a *set-2* catalytic inactive allele is not sufficient to restore proper PVQ development, indicating that SET-2, but not SET-16, is required for the heat-induced PVQ phenotype (Fig 4F). It should be noted that the mutation in *set-2* is not dominant, as young *set-2* males (grown at 20°C) crossed with young WT hermaphrodites (grown at 20°C) does not result in PVQ defects in the F1 cross-progeny (mean ± sem: 6% ± 0.7%, n = 150), therefore suggesting that a reduced SET-2 activity sensitizes the PVQ neurons to environmental effects.

It is challenging to discriminate the influence of parental-contributed inactive SET-2 protein from the parental-contributed aberrant chromatin landscape, generated by and inherited along with *set-2* mutation, on the intergenerational epigenetic phenotype observed under certain parental conditions. However, the fact that the cross-progeny of old WT males with old WT hermaphrodites displays PVQ defects despite carrying two WT *set-2* alleles, suggests that the aberrant posttranslational modifications in chromatin itself contribute to the intergenerational inheritance of the PVQ phenotype.

Altogether, these results emphasize the relevance of maintaining proper H3K4me3 levels across generations and the critical role of SET-2 in preventing deleterious inter and transgenerational effects in nervous system homeostasis (Fig 4G).

## Discussion

In this study, we show that aberrant H3K4 methylation related to genetic mutations in H3K4 regulators and to certain conditions can influence the development of the nervous system across generations.

We first show that perturbation of H3K4me3, by inactivating *set-2* or ablating *ash-2* in a parental generation, can have transgenerational effects on neuronal development, resulting in aberrant PVQ axon guidance in genetically WT descendants. ChIP and RNA sequencing analyses strongly suggest that this inheritance pattern is related to the fact that parental inactivation of *set-2* decreases H3K4me3 levels and impacts transcription for several generations, independently of the genotype. These results indicate not only that an ancestral H3K4me3 status can be transmitted with

influence on transcription in the following generations, but also that the normal pattern of H3K4me3, when perturbed, is slowly reestablished, creating a window of epigenetic instability and vulnerability that can affect neuronal development and could increase the sensitivity to endogenous and exogenous cues. The transgenerational effect of perturbed H3K4me3 is also testified by the fact that homozygous *set-2* mutations must be carried over multiple generations before causing a fully penetrant PVQ phenotype.

Second, we show that PVQ axon guidance is sensitive to physiological (age) and environmental (temperature) conditions and importantly, SET-2 contributes to this susceptibility acting as a downstream effector responsible for the PVQ defects and for the aberrant transcription observed in these conditions. In aged animals or animals experiencing a rise of temperature, we noted an overall increase of H3K4me3, correlating with transcription upregulation. This effect agrees with a study showing that high temperature results in a more open chromatin state (Rogers & Phillips, 2020). At first glance, this result is in contrast with the observation that reduced H3K4me3 levels in *set-2* mutants and outcrossed progeny are responsible for the transmission of the PVQ phenotype. One way to reconciliate this evidence is to consider the conditions (advanced maternal age and temperature) and *set-2* mutations as acting in opposite directions with respect to H3K4 methylation but having a similar consequence in neuron development. In support of this possibility, animals with increased H3K4me3 level, as the ones defective for the H3K4me2/3 demethylase *rbr-2*, present a similar PVQ phenotype (Mariani et al, 2016), indicating that higher or lower levels of H3K4me3, compared with WT, are equally deleterious for the PVQ development. We therefore believe that H3K4me3 levels must be finely established and maintained in a tight equilibrium by the action of H3K4me3-regulating enzymes between generations to achieve proper neurodevelopment, both in standard and abnormal conditions.

It should be noted that although a strong reduction of H3K4me3 (as in *set-2* catalytic-inactive mutants) has a true transgenerational effect on PVQ development, spanning at least three generations, the increased level of H3K4me3 observed in aged parents or in animals exposed to high temperature has intergenerational effects, lasting for only one generation. This evidence suggests that to achieve transgenerational effects, changes in H3K4 levels must be complemented by other "factors," not involved in the intergenerational inheritance. It is probable that these other factors are of epigenetic nature, as suggested from a study addressing the mechanism of transgenerational inheritance of longevity (Lee et al, 2019) observed in mutant animals for WDR-5, a component of the COMPASS H3K4 methyltransferase complex. Here, the authors show that the reduction of H3K4me3 levels associated to *wdr-5* loss is accompanied by increased H3K9me2 across generations and that

---

defects in self-progeny from 4DOA WT hermaphrodites and in cross-progeny from hermaphrodites crossed with males, in different age combinations. n ≥ 190 pr. condition. \*\*$P < 0.01$, \*\*\*$P < 0.001$, n.s. not statistically significant, comparing with self-progeny from 4DOA hermaphrodites. **(E)** PVQ defects in cross-progeny from 4DOA hermaphrodites crossed with L4 males either WT, *set-2(zr1208)* or *set-16(zr1804)*. n ≥ 150 pr. condition. \*\*$P < 0.01$, n.s. not statistically significant, comparing with self-progeny from 4DOA hermaphrodites. **(F)** PVQ defects in cross-progeny from hermaphrodites kept at 25°C for two generations crossed with males kept at 20°C until adulthood (crossing done at 25°C) either WT, *set-2(zr1208)* or *set-16(zr1804)*. n ≥ 150 pr. condition. \*\*$P < 0.01$, n.s. not statistically significant, comparing with the self-progeny from 4DOA hermaphrodites. In (A, B, D, E, F), data are expressed as mean ± SEM. Each black dot represents an independent experiment. Statistical significance was assessed using one-way ANOVA Tukey's multiple comparison. **(G)** Schematic model for H3K4me3 contribution in neuronal homeostasis across generations.

the transgenerational longevity observed in *wdr-5* mutant animals requires the H3K9 methyltransferase MET-2. Thus, further analyses revealing chromatin changes when H3K4me3 is lost in *set-2* catalytic inactive mutants and when H3K4me3 is increased in animals experience temperature increase and aging are required to understand and distinguish the mechanisms behind inter and transgenerational inheritance. Nevertheless, our results reinforce the notion (Greer et al, 2011; Kelly, 2014) that H3K4me3 is a memory mark that is not fully resettled over generations and, more in general, that histone modifications are involved in intergenerational responses to particular conditions and their loss, in a parental generation, can have a transgenerational impact.

Next-generation sequencing of human samples provides strong evidence that H3K4 regulators are critical for normal nervous system development and are considered causative of neurodevelopmental diseases, when mutated. This study demonstrates that incorrect deposition of H3K4me3 has consequences for the fine structure of the nervous system over multiple generations. As the epigenetic machinery is conserved, it is possible that the same principles apply in mammals and our findings might therefore provide insights related to inheritance patterns of neurodevelopmental diseases.

# Materials and Methods

### Genetics and strains

*C. elegans* strains were cultured using standard growth conditions at 20°C with *Escherichia coli* OP50 (Brenner, 1974) unless otherwise stated. The *set-2(zr1208)* and *set-2(zr2012)* were backcrossed four times to N2. Neuronal marker strains were backcrossed at least three times to N2 before the analyses. Strains used are reported in Table S3.

### CRISPR lines

The CRISPR line *set-2(zr1208)* was created by injecting ssDNA repair template (CTACGCGATGGAGTCGATTGCACCAGATGAGATGATTGTCGAAT-TCATCGGACAGACGGTCAGTTTTTTTTGTGAAATTAAATTCCGAA) for *set-2* and desired guide RNA cloned into pJJR50 (*zr1208* sgRNA sequence AGATGATTGTGGAGTATAT). The mix also contained a *pha-1* repair template and pJW1285 (driving expression of Cas9) (Ward, 2015). The mix was injected into *pha-1(e2123)* mutants. All constructs were injected at a concentration of 50 ng/µl. Selection for *pha-1* WT clones was performed at 25°C. Mutations were confirmed by sequencing. The mutation in the *zr1208* allele was selected based on the following criteria: (1) conserved from yeast to humans; (2) sitting in the catalytic pocket reducing methyltransferase activity in MLL1 based on (Southall et al, 2009) (3) a conservative substitution (Y1397 to F). The line *set-2(zr2012)* was created using the same strategy (Abay-Nørgaard et al, 2020).

### Outcrossing procedure

The crossing procedure in Fig 1B was performed as outlined here. Seven young N2 L4 males was incubated on plates with four L4 hermaphrodites of desired genotypes (including *oyIs14[Psra-6::GFP]* to visualize PVQs). After 3–4 d, F1 L4 progeny was singled out. Plates with cross-progeny were identified by the lack of *oyIs14* marker in ~25% of the F2 progeny. The L4 F2 progeny was singled out and left on plates for 2 d and then genotyped to identify F3 homozygous mutant and WT descendants. In experiments where the following generations were examined, five L4 hermaphrodites were moved to new plates between each generation. Each crossing procedure was performed at least four times when scoring PVQs in WT descendants. One WT-des line from *set-2(zr2012)* was lost between F3 to F4 because of incubator meltdown. After mutant descendants in *set-2(zr1208)*, the crossing procedure was performed two times. For RNA-seq and ChIP-seq experiments, descendants originated from the same cross. All mutants were kept in culture for at least 10 generations before performing the cross. The PVQ defects in *set-2(zr1208)* after 10 generations were 27% ± 3.1% (n ≥ 150).

### Environmental changes

Heat: Five worms L4 stage were moved from 20°C to 25°C, and between each generation, five worms in the L4 stage were moved to new plates. The progeny used in all heat experiments were always from 1-d-old hermaphrodites.

Crossing with non-heat-exposed males were performed as follows. Seven F2 L4 hermaphrodites grown at 25°C were crossed (at 25°C) with five *oyIs14* L4 males grown at 20°C. F3 cross-progeny was identified by the presence of *oyIs14*.

Maternal age: 50 WT worms L4 were moved to a single plate and moved to new plates every 24 h (second plate was considered as the progeny from 1-d-old adults). The progeny from 3-d-old adults was produced at 96–120 h after L4 (fourth plate) and the progeny from 4-d-old adults was produced at 120–144 h after L4 (fifth plate).

Crossing with males and hermaphrodites of different ages were performed as follows. Ten 4-d-old hermaphrodites were crossed with either 10 young or thirty 4-d-old *oyIs14* males and five young hermaphrodites were crossed with 34-d-old *oyIs14* males. Cross-progeny was identified by the presence of *oyIs14*.

Starvation: 10 L4 worms were added to a single well (96-well flat bottom plate) with 200 µl of M9. After 24 h, worms were moved back to standard NGM plates and left to produce progeny for 24 h.

### Western blot

Protein extracts were always prepared from L4 animals (except when comparing 1-d-old adult and 4-d-old adults). Samples were boiled in SDS–PAGE buffer for 2 × 5 min and sonicated for 10 min using a Diagenode Bioruptor (UCD-300). The following antibodies were used: anti-H3K4me3 (C42D8; 1:750; Cell Signaling Technology); anti-H3K4me1 (ab8895; 1:1,000; Abcam); anti-H3K4me2 (07-030; 1:1,000; Millipore) anti-H3 (ab1791; 1:10,000; Abcam); anti-H3K9me3 (ab8898, 1:1,000; Abcam); anti-H3K27me3 (MABI0323, 1:1,000; Thermo Fisher Scientific); anti-H3K36me3 (D5A7; 1:1,000; Cell Signaling Technology); anti-H3K79me2 (D15 × 10$^8$; 1:1,000; Cell Signaling Technology) and peroxidase-labelled anti-rabbit and anti-mouse

secondary antibodies (1:10,000; Vector Laboratories). Western blots were quantified using ImageJ (National Institutes of Health).

### Neuronal scoring

Worms were immobilized in sodium azide and placed on microscope slides with a 5% agarose pad. PVQs were scored at the L3 to L4 stages. All other neurons were scored at the L4 to YA stages. WT, coming from young mothers (maximum 48 h after the L4 stage), were grown for at least five generations at 20°C. Micrographs were obtained using a Zeiss AXIO imager M2 fluorescence microscope.

### Statistical analyses

Graphpad prism 9 was used for all statistics on neuronal phenotypes. For testing statistical significance for each independent line after outcrossing, we used chi-square test and/or Fisher's exact test. For every line, we scored between 50–52 animals and compared with WT (6%, n = 200). Both chi-square test and/or Fisher's exact test predict that if the penetrance is higher or equal ~16% (8 out 50–52 animals with defective PVQs) the $P$-value is less than 0.05. All other neuronal scoring were tested using one-way ANOVA with Tukey's multiple comparison or $t$ test.

### RNA sequencing and analysis

RNA was isolated from three to four independent experiments during heat treatment and maternal age. Every sample contained 60–75 mid-L4 animals. During outcrossing procedure, RNA was isolated from three to four independent lines. Every sample contained 25–30 mid-L4 animals. The mid-L4 stage was chosen for synchronization purposes and to acquire enough RNA. RNA was extracted using an Arcturus PicoPure RNA Isolation Kit (KIT0204; Thermo Fisher Scientific). Sequencing libraries were constructed using a TruSeq RNA Library Prep Kit v2 (RS-122-2001/2; Illumina). Libraries were sequenced using a NextSeq 500 system and a NextSeq 500/550 High Output Kit v2 (Illumina).

RNA sequencing results were analysed using Galaxy (v22.01.1.dev0). FastQC (v.0.11.9) was performed to test read quality. Reads were trimmed with Trimmomatic (v.0.38), Illuminaclip (Truseq3 single-end, default), crop (73). RNA star (2.7.8a) was used to align reads. Htseq-count(0.9.1) was used to count mapped reads to the C. elegans genome (WS220). DESeq2 (v1.34.0) was used to determine differentially expressed genes (DEGs) and to generate PCA plots for the environmental data (Figs 3F and G and 4C).

Principle component analysis was conducted with Python package skikit-learn for transcriptomic data across generations (Fig 2E and F). DEGs were acquired by comparing each group (wt-ctrl, WT-des, and MUT-des) using EdgeR (Robinson et al, 2010). The DEGs were clustered using hierarchical clustering implemented in R. Ward's criterion for genes with 1—(correlation coefficient) was used as a distance measure. A clustering heatmap was drawn using a z-score that is scaled across samples for each gene. Genes with a false discovery rate < 0.05 and $\log_2$fold change > 1.0 were selected as DEGs.

For outcrossing and environmental procedures, aligned reads were between 21.2 M and 13.6 M pr. sample on average, respectively.

### ChIP sequencing and analysis

Every chromatin sample consisted of exactly 100 L4 animals (mix of two independent lines, 50 animals each). The samples were washed two times in M9 (in protein low-bind tubes). The samples were left on a shaking table for 1 h to purge worm guts from bacteria, followed by two washes in M9. The M9 was aspirated until 400 $\mu$l was left and 100 $\mu$l of 10% PFA (methanol-free) was added to the samples followed by 30 min rotation at RT. 55 $\mu$l of Tris–HCL (pH 7.5) was added to quench reaction. Worms were washed three times in ice-cold M9 and spun down at max speed for 1 min (tabletop centrifuge) between each wash. All liquid was aspirated, and the samples were kept at –80°C until further processing.

Spike-in ChIPseq was carried out as previously described Godfrey et al (2019) with minor modifications. Briefly, 100 fixed worms were used as the starting material and mouse embryonic stem cells were used for spike-in. The samples were sonicated using a Covaris E220evolution to 100–500 bp. DNA was purified using QIAGEN minElute PCR purification columns and DNA libraries were generated using the NEB Next Ultra DNA library preparation kit for Illumina (Cat no. E7370). The samples were sequenced by 41 bp paired-end reads on the Illumina NextSeq 500 platform. 1 $\mu$l H3K4me3 antibody (same as for western blot procedure) was used per ChIP reaction.

For Spike-IN ChIPseq, paired-end reads were processed with trim_galore (Martin, 2011) followed by mapping to the ce11 and mm10 genomes using bowtie1/1.1.2 (Langmead et al, 2009). PCR duplicates were removed from uniquely mapping reads using samtools/1.10 (Li et al, 2009). The spike-in normalization factor was derived as previously described Orlando et al (2014) and further corrected using the ratio of mm10/ce11 total read counts in the corresponding inputs (Fursova et al, 2019). Peaks were called for each sample using macs2/2.1.1.20160309 (Zhang et al, 2008) and further processed using bedtools/2.30.0 (Quinlan & Hall, 2010). In each condition, peaks found in all three biological replicates were used for further analysis. Peaks found in all conditions were combined to generate a union of peaks—"All-peaks" or intersected to generated "Common-peaks." deeptools/3.2.1 (Quinlan & Hall, 2010) was used to generate Spike-In-normalized BigWigs using a bin size of 10 bp while excluding blacklisted regions (ce11.v2) (Amemiya et al, 2019). These were used for downstream analysis to generate enrichment, PCA, and correlation plots using deeptools/3.2.1 (Quinlan & Hall, 2010). BigWigs were visualized using IGV/2.13.0 (Robinson et al, 2011). ChIPseq samples and publicly available data were processed in the same manner with RPKM normalization used instead and two biological replicates for the H3K4me3 ChIPseq in L4 stage larvae.

### Material and Correspondence

For data request, correspondence should be addressed to Anna Elisabetta Salcini or Steffen Abay-Nørgaard.

## Data Availability

RNA-seq and ChIP-seq data are in GEO (https://www.ncbi.nlm.nih.gov/geo/), Gene Expression Omnibus Accession GSE215921. The following

publicly available dataset was used: H3K4me3 ChIPseq in L4 adult worms (Jänes et al, 2018), Gene Expression Omnibus Accession GSE114440.

# Supplementary Information

# Acknowledgements

We thank theCaenorhabditis Genetics Center, which is funded by the NIH Office of Research Infrastructure Programs (P40 OD010440); the National BioResource Project for *C. elegans* (Japan) and the International C. elegansGene Knockout Consortium for providing strains. We thank Biorender.com, which was used to produce figures. We thank Kristian Helin (ICR, London) for comments on the manuscript. This work was supported by Lundbeckfonden (project ref. R324-2019-1452) to S Abay-Nørgaard. Work in the Porse laboratory was supported through a grant from the Novo Nordisk Foundation (Novo Nordisk Foundation Center for Stem Cell Biology, Dan-Stem; Grant Number NNF17CC0027852).

## Author Contributions

S Abay-Nørgaard: conceptualization, data curation, formal analysis, funding acquisition, and writing—original draft.
MC Tapia: data curation, formal analysis, methodology, and writing—original draft.
M Zeijdner: data curation and formal analysis.
JH Kim: data curation and formal analysis.
KJ Won: data curation, formal analysis, and supervision.
B Porse: data curation and supervision.
AE Salcini: conceptualization, data curation, supervision, project administration, and writing—original draft, review, and editing.

## Conflict of Interest Statement

The authors declare that they have no conflict of interest.

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
