## [Reviewer comments · Life Science Alliance]

Life Science Alliance

Inter- and trans-generational impact of H3K4 methylation in neuronal homeostasis

Steffen Abay-Nørgaard, Marta Tapia, Mandoh Mandoh Zejdner, Jeonghwan Kim, Kyoung Won, Bo Porse, and Anna Elisabetta Salcini

DOI: <https://doi.org/10.26508/lsa.202301970>

Corresponding author(s): Anna Elisabetta Salcini, University of Copenhagen and Steffen Abay-Nørgaard,

Review Timeline:

Submission Date:	2023-02-05
Editorial Decision:	2023-03-16
Revision Received:	2023-04-26
Editorial Decision:	2023-05-12
Revision Received:	2023-05-16
Accepted:	2023-05-17

Transaction Report:

March 16, 2023

Re: Life Science Alliance manuscript #LSA-2023-01970-T

Dr. Anna Elisabetta Salcini
University of Copenhagen
BRIC
Ole Maaløes vej 5
Copenhagen 2200
Denmark

Dear Dr. Salcini,

Thank you for submitting your manuscript entitled "Inter- and trans-generational impact of H3K4 methylation in neuronal homeostasis" to Life Science Alliance. The manuscript was assessed by expert reviewers, whose comments are appended to this letter. We invite you to submit a revised manuscript addressing the Reviewer comments.

Thank you for this interesting contribution to Life Science Alliance. We are looking forward to receiving your revised manuscript.

Sincerely,

B. MANUSCRIPT ORGANIZATION AND FORMATTING:

Reviewer #1 (Comments to the Authors (Required)):

This study addresses a current topic of interest to the field of epigenetics: whether the inheritance of a histone modification can affect functional phenotypes in subsequent generations. Specifically, the authors investigate whether the active histone modification H3K4me3 has an effect on individuals in the F1 generation (via inter-generational inheritance) or in the F3 generation and beyond (via trans-generational inheritance). They are using a *C. elegans* axon guidance model established in a previous study: mutants lacking the activity of H3K4me modifiers have development defects specifically in PVQ interneurons. In this study, the authors examine whether the PVQ defective phenotype, along with associated changes in H3K4me levels and gene expression, can be inherited by wild-type descendants. Interestingly, the authors find that wild-type F3 (and sometimes F4) descendants of mutants lacking the activity of COMPASS complex components *set-2* or *ash-2* inherit the PVQ defect. In support of their model, they also demonstrate that these F3 wild-type descendants have correspondingly low H3K4me3 levels and misregulated gene expression. They next use their PVQ model to test whether environmental factors can affect heritability, and implicate SET-2 in the intergenerational inheritance of defects caused by temperature stress or maternal age. Overall, this study demonstrates that levels of H3K4me3 correlate with the epigenetic inheritance of a PVQ neuronal defect for up to four generations, and establishes a model for understanding how heritable chromatin modifications affect neurodevelopment.

Major points of the paper:

1. The data presented in Figure 1 strongly support the transgenerational inheritance of the PVQ defect.
 - a. However, it didn't seem like these data were summarized in a table. (The supplementary files weren't made available to reviewers, so I wasn't able to fully confirm the absence of a summary table, but no mention was made of one in the text.) The authors should include a summary table for each individual replicate, clearly stating how many animals were WT vs. PVQ-defective in 1. each individual line, 2. each generation examined, and 3. For each transgenerational experiment. Given the stochastic nature of transgenerational inheritance seen in previous studies (for a range of phenotypes), these data will help demonstrate the intrinsic variability of the effect on PVQ guidance. Furthermore, this table could also be used to indicate what comparisons were made to identify statistical significance, and represent the p-values of those comparisons.
2. The data presented in Figure 2 strongly support the transgenerational nature of the decrease in genomic H3K4me3 levels, and do also indicate the heritable effect on gene expression. However, these genomic analyses need to be further developed to better support the conclusions drawn.
 - a. For the H3K4me3 ChIP-seq, it is interesting that WT descendants clustered with some overlap between F3 and F5 in Fig 2A. This would may also correspond with the incomplete penetrance of the PVQ defect and the gradient effect seen in the RNA-seq of the F3 WT descendants. Are the authors able to determine whether the F3 sample that more closely resembled the F5 samples had a less severe PVQ defect (in the two lines used for the sample)?
 - b. Additionally, For Figure 1B, C, and D, to better support their claim that levels of H3K4me3 are decreased in F3 compared to F5 wild-type descendants, the authors should discuss whether there were any peaks shared between these two populations. Were the F3 peaks unique to those in the F3 generation, or were most of the F3 peaks also found in the F5 generation? It is formally possible that the increase in overall H3K4me3 levels is driving solely by H3K4me3 arising at new peaks in F5 (ie: as gene expression returns to more WT levels). Therefore, a metaplot comparing H3K4me3 levels only at shared peaks would help distinguish between this possibility, or further support the authors' conclusion that both the amount of H3K4me3 at each locus, in addition to the number of genomic loci, may account for the return to a wild-type PVQ phenotype.
 - c. The RNA-seq analysis felt very cursory, and the heatmap shown in Figure 1F makes it difficult to fully parse the authors conclusions. For example, it's hard to tell from the heatmap, whether F3 wild-type descendants have more up-regulation compared to the wild-type control, or do they have more down-regulation (which is what would be expected given the active nature of H3K4me3)? More computational analysis would help strengthen these conclusions: the authors should split genes into an up-regulated class and a down-regulated class, then show a dot plot of relative gene expression levels for each of those classes in wild-type F3 and F5 descendants. This dot plot would allow them to better demonstrate their claim (P8 line 22) that the F5 look more similar to the WT control than to the mutant descendants.
3. Although Figure 3 reports interesting observations about the effect of heat and maternal age on their PVQ model, the discussion of the reasoning behind these experiments was lacking, and some conclusions did not fully consider alternatives.

- a. Both the results and the discussion fail to mention the studies showing how temperature affects chromatin state with another histone modification, H3K9me (Ni et al 2016, Delaney et al 2019, and Costello & Petrella 2019).
- b. I agree that there is a difference in the PVQ defect observed between F1 and F2/F3 generations maintained at 25C, but disagree with the interpretation that penetrance of this defect plateaus at F3 - it appears to reach its maximum effect by F2. Therefore, this result could simply represent a maternal effect (ie: intergenerational) of the parental population being raised at 20C until L4 stage (with gonad development and spermatogenesis occurring at the control temperature).
- c. I found it difficult to follow the relationship between the effects of heat/maternal age, the heritability of the PVQ defect, and overall H3K4me3 levels. The elevated H3K4me3 levels in animals raised at 25C could just be due to a heat shock response, while the elevated H3K4me3 levels in older adults could be age-related changes in up-regulation of gene expression (as suggested by Pu et al 2018). This interpretation is further supported by the authors' observation that the only other histone modification that changed in aged hermaphrodites was H3K36me3, which is closely associated with active transcription. Additionally, although the difference in H3K4me3 levels observed in L4-stage F1 progeny from aged mothers is compelling, but doesn't in itself indicate that the F1 progeny necessarily inherited high levels of H3K4me3. The increased levels of H3K4me could just be an altered response to other defects that come from aged oocytes (like the nutritional differences in these oocytes reported by Perez et al 2017).

4. The data presented in Figure 4 do nicely demonstrate that, since the authors did not observe an additive effect in the absence of SET-2 activity for either heat or maternal age, SET-2 is likely responsible for the PVQ defects caused by these environmental exposures.

- a. The relationship between SET-2 (and therefore, with H3K4me3 itself) would be strengthened further if they could show that a set-2 mutation is haploinsufficient only in the presence of heat or advanced maternal age - this experiment would help indicate that removing half of SET-2's activity can sensitive PVQ neurons to environmental effects.
- b. I found the results shown in Figure 4D-F very interesting, as these data indicate whether the epigenetic effect is in cis (ie: mediated by the histone modification itself) or could be a parental effect in trans (ie: mediated by the M-Z+ or P-Z+ contribution of SET-2 enzyme) - interpreting these results should be added to the discussion, as they may indicate the mechanism of intergenerational inheritance.

Minor issues:

1. The introduction nicely establishes the research model and summarizes the status of the field. However, to provide a more complete context to understand the major points of this study, it should include more discussion of the following points:
 - a. It will be important for the authors to fully define inter-generational inheritance versus trans-generational inheritance. This distinction is an important difference in observations presented in this study, and becomes important for interpreting their findings, since the molecular mechanisms underlying each type of inheritance are likely to be distinct.
 - b. Page 3 line 15: The phrasing of "grandparental age or tobacco consumption matters in case of neurodevelopmental disorders in grandchildren" is imprecise in a way that obscures the nature of the data. The use of "matters" implies causation. However, because these citations refer to human epidemiological studies, this should be reworded to accurately reflect the correlative relationship of these studies.
 - c. For the statement on page 4 line 4, the authors should also mention work implicating the importance of H3K4 KDM LSD1/KDM1A in mouse neurodevelopment, particularly because these studies address the potential maternal effect for neurodevelopmental defects in Kdm1a M-Z+ individuals (Ancelin et al 2016 and Wasson et al 2016)
 - d. Page 4 line 15: when discussing the role of H3K4me in heritable epigenetic phenotypes, the authors did not include any mention of the first demonstrated instance of transgenerational inheritance of H3K4me (affecting fertility as shown in Katz et al 2009 and Nottke et al 2011) and failed to include discussion of SPR-5's role in longevity, as demonstrated in Greer et al 2016.
2. I found the abbreviations "wt des" and "mut des" difficult to read, in part because they look almost like words. Replacing these with "WT-des and MUT-des", would help readers parse the text a little more easily.

3. Typos:

- a. Page 17 line 3 - "genotyped two identify" should be "to"
- b. P6 line 7 - "descendants carry wild-type alleles" should be "carrying"
- c. P9 line 5 - "guidance hinge on inherited H3K4me3 level" should be "hinges" and "levels"

Reviewer #2 (Comments to the Authors (Required)):

In figure 1 the authors show that a PVQ axon guidance defect in set-2 mutants can be inherited. They also show that it is not HRDE-1 dependent. In figure 2 the authors show that there are inherited H3K4me3 and transcriptional changes in descendants of set-2 mutants. In figure 3, the authors show that temperature and maternal age effect PVQ and other neuron axon guidance, along with corresponding increases in transcription, but starvation does not have an effect. Also, the effects are not inherited in subsequent generations. In figure 4, the authors show that the increased axon guidance effect at elevated temperature and with advanced maternal age is dependent upon set-2, but not set-16. Finally, the authors show that having a young male or young hermaphrodite in a cross, suppresses the effect of advanced parental age. Also, crossing from a male at normal temperature

suppressed the effect of elevated temperature. In both cases, this suppression is dependent upon set-2, but not set-16.

Overall, this paper has some really interesting observations in a number of different paradigms. It is perhaps somewhat disappointing that some of the most interesting observations are not really followed further (see below). There is also, I believe, a bit of lack of clarity in how the results are presented compared to one another. Specifically, some of the experiments are clearly opposite one another and should be presented that way. Finally, I think there is a missed opportunity for further conclusion (see below). Nevertheless, the broad survey presented here of observations related to manipulation of H3K4 methylation are very interesting and will be of broad interest to range of fields. As a result, with some changes to the presentation, I am overall very enthusiastic about this manuscript.

Overall in figure 1, the effect is somewhat modest. The authors only observe a significant inheritance effect in 7 out of ~40 lines through the hermaphrodite and only 2 when inheritance occurs through the male. Overall, the inheritance through the hermaphrodite at the population level (1D) does look different than the WT (1G), particularly because the overall % of PVQ defects is higher, but the inheritance through the male S1C, does not look different to me than WT (1G). In fact, the average looks to me to be lower upon inheritance through the male (S1C) than WT (1G). The authors need to do population level statistics to determine if there is any difference between 1G vs. S1C, or even 1G vs. 1D, 1E and 1F. For example, it would make some sense if S1C is not different than 1G, that the inheritance can only occur through the oocyte. The authors should also report the average % defect for each, so the comparison can be more directly made. It would also be nice if the authors reported in the text, the exact number of lines with a statistically significant PVQ defect out of the total number of lines assayed.

The control in figure 1 showing that the PVQ defect is not inherited in a mutant with a known PVQ defect that is not epigenetically derived is very nice.

In mutdes of set-2 mutants the authors observe that the vast majority of lines at F3 no longer have a significant guidance defect, but the defect returns in all lines by F6. This suggests that it requires multiple generations of being homozygous mutant for set-2 to observe that guidance effect and that it is actually a transgenerational effect itself. The authors sort of suggest this in the discussion, but this interpretation could be clearer. In any case, this is one of the more interesting observations from the manuscript, that is not really followed up. Particularly since a similar observation has been made for the COMPASS complex with regard to longevity (Lee et al eLife 2019). Could the effect here also be due to K9? The authors should definitely discuss this possibility.

In wt des, H3K4me3 increased from F3 to F5. This is reminiscent of Greer et al Nature 2011, where lifespan remains extended in descendants of compass mutants, despite normal compass which might be expected to restore normal H3K4me3 levels immediately. Could the increase in H3K4me3 levels be necessary for the return to normal lifespan? One way to look at this is to ask whether the increase in H3K4me3 from F3 to F5 is necessary to restore WT H3K4me3. This is certainly suggested by the RNAseq which returns towards WT in F3 and F5 wt des. But is it known that the increase in H3K4me3 in wt des between F3 and F5 returns it more towards normal? The authors probably should have compared H3K4me3 in wt des compared to WT? Perhaps this could be assessed by comparing to published H3K4me3 in L4s? Also, is it possible that the increase in H3K9 observed in Lee et al is what temporarily restricts full H3K4me3 in wt des at F3? The authors could look at this by ChIP or even western blot.

On a technical note, in 2C it would be nice to plot (perhaps in the supplement) each F3 vs the corresponding F5 to see if H3K4me3 is always increasing in each individual replicate of the experiment.

Rogers and Phillips NAR 2020 showed that elevating temperatures opens chromatin, which would be the opposite of mutating set-2. This is consistent with what the authors found in 3E. Therefore, the more interesting question might be to reduce temperature and see if it recapitulates what happens in set-2 mutants? The authors do show that elevating temperature in set-2 mutants suppresses the effects of elevated temperature, which is consistent with elevated temperature and set-2 mutation acting in opposite directions. But overall there seems to be a lack of clarity with respect to the effect on the axon guidance. At times the authors seem to imply that set-2 mutants, elevated temperature and advanced maternal age are the same because of the similarity in effect on the axon guidance phenotype. But the opposite molecular effects and the suppression clearly indicate that they are different. The main point, it seems to me, is that modulating chromatin transcription in both directions causes the same phenotype. This ultimately probably the most significant conclusion of the paper, but is somewhat buried. The authors ultimately arrive at this conclusion in the discussion, but the presentation in the results is somewhat confusing. Also, another really important conclusion from this work that is not really emphasized is that, despite both directions affecting axon guidance, there is only epigenetic inheritance of this defect when chromatin and gene expression is decreased in a set-2 mutant. This is perhaps consistent with the data from Lee et al suggesting that the inherited effect is not due to the lack of H3K4me3, but rather due to the corresponding increase in H3K9 methylation. In my opinion, the paper would be much more impactful if these interpretations are more emphasized in the abstract, discussion and perhaps even the title.

We thank the reviewers for the positive and thoughtful comments that helped us improving the quality of our manuscript. We are encouraged that they found our results very interesting and of broad interest to range of fields.

Changes to the main text are highlighted in red in the revised manuscript.

Reviewer #1 (Comments to the Authors (Required)):

This study addresses a current topic of interest to the field of epigenetics: whether the inheritance of a histone modification can affect functional phenotypes in subsequent generations. Specifically, the authors investigate whether the active histone modification H3K4me3 has an effect on individuals in the F1 generation (via inter-generational inheritance) or in the F3 generation and beyond (via trans-generational inheritance). They are using a *C. elegans* axon guidance model established in a previous study: mutants lacking the activity of H3K4me modifiers have development defects specifically in PVQ interneurons.

In this study, the authors examine whether the PVQ defective phenotype, along with associated changes in H3K4me levels and gene expression, can be inherited by wild-type descendants. Interestingly, the authors find that wild-type F3 (and sometimes F4) descendants of mutants lacking the activity of COMPASS complex components *set-2* or *ash-2* inherit the PVQ defect. In support of their model, they also demonstrate that these F3 wild-type descendants have correspondingly low H3K4me3 levels and misregulated gene expression. They next use their PVQ model to test whether environmental factors can affect heritability, and implicate SET-2 in the intergenerational inheritance of defects caused by temperature stress or maternal age.

Overall, this study demonstrates that levels of H3K4me3 correlate with the epigenetic inheritance of a PVQ neuronal defect for up to four generations, and establishes a model for understanding how heritable chromatin modifications affect neurodevelopment.

Major points of the paper:

1. The data presented in Figure 1 strongly support the transgenerational inheritance of the PVQ defect.
a. However, it didn't seem like these data were summarized in a table. (The supplementary files weren't made available to reviewers, so I wasn't able to fully confirm the absence of a summary table, but no mention was made of one in the text.) The authors should include a summary table for each individual replicate, clearly stating how many animals were WT vs. PVQ-defective in 1. each individual line, 2. each generation examined, and 3. For each transgenerational experiment. Given the stochastic nature of transgenerational inheritance seen in previous studies (for a range of phenotypes), these data will help demonstrate the intrinsic variability of the effect on PVQ guidance. Furthermore, this table could also be used to indicate what comparisons were made to identify statistical significance, and represent the p-values of those comparisons.

We thank the reviewer for raising this important point. We now summarize the data in Fig. S1G and present the detailed results in Table S1, as requested.

2. The data presented in Figure 2 strongly support the transgenerational nature of the decrease in genomic H3K4me3 levels, and do also indicate the heritable effect on gene expression. However, these genomic analyses need to be further developed to better support the conclusions drawn.
a. For the H3K4me3 ChIP-seq, it is interesting that WT descendants clustered with some overlap between F3 and F5 in Fig 2A. This would may also correspond with the incomplete penetrance of the PVQ defect and the gradient effect seen in the RNA-seq of the F3 WT descendants. Are the authors

able to determine whether the F3 sample that more closely resembled the F5 samples had a less severe PVQ defect (in the two lines used for the sample)?

We thank the reviewer for the interesting observation. Unfortunately, the correlation analysis suggested is not possible, because the low number of synchronized L4 animals produced from a single F2 mother.

b. Additionally, For Figure 1B, C, and D, to better support their claim that levels of H3K4me3 are decreased in F3 compared to F5 wild-type descendants, the authors should discuss whether there were any peaks shared between these two populations. Were the F3 peaks unique to those in the F3 generation, or were most of the F3 peaks also found in the F5 generation? It is formally possible that the increase in overall H3K4me3 levels is driving solely by H3K4me3 arising at new peaks in F5 (ie: as gene expression returns to more WT levels). Therefore, a metaplot comparing H3K4me3 levels only at shared peaks would help distinguish between this possibility, or further support the authors' conclusion that both the amount of H3K4me3 at each locus, in addition to the number of genomic loci, may account for the return to a wild-type PVQ phenotype.

We believe the reviewer refers here at Fig. 2B, C and D. We thank the reviewer for this suggestion and agree that this analysis is important. Peak overlap between F3 and F5 wild-type descendants shows that the majority of F5 peaks is found in the F3 generation (Fig.S2D) and H3K4me3 signal is increased in the F5 generation compared to F3 at these peaks (Fig.S2C).

c. The RNA-seq analysis felt very cursory, and the heatmap shown in Figure 1F makes it difficult to fully parse the authors conclusions. For example, it's hard to tell from the heatmap, whether F3 wild-type descendants have more up-regulation compared to the wild-type control, or do they have more down-regulation (which is what would be expected given the active nature of H3K4me3)?

Differentially expressed genes are equally divided in up- and down-regulated genes. While not expected considering the “activating” nature of the H3K4me3, this result is in line with previous observations reporting that loss of H3K4me3 results in both up- and down-regulation of gene expression (e.g. Abay-Norgaard et al, 2020).

More computational analysis would help strengthen these conclusions: the authors should split genes into an up-regulated class and a down-regulated class, then show a dot plot of relative gene expression levels for each of those classes in wild-type F3 and F5 descendants. This dot plot would allow them to better demonstrate their claim (P8 line 22) that the F5 look more similar to the WT control than to the mutant descendants.

We generated the analysis requested (see below). Probably due to the small number of gene up- and down-regulated (49 and 41, respectively), the plots do not help clarify the point suggested by Fig.2E and 2F, that F5 lines look more like to WT compared to F3 lines.

3. Although Figure 3 reports interesting observations about the effect of heat and maternal age on their PVQ model, the discussion of the reasoning behind these experiments was lacking, and some conclusions did not fully consider alternatives.

We apologize for the lack of clarity in this part of the manuscript. We now change and correct the text to better explain the reasoning behind these experiments.

a. Both the results and the discussion fail to mention the studies showing how temperature affects chromatin state with another histone modification, H3K9me (Ni et al 2016, Delaney et al 2019, and Costello & Petrella 2019).

We thank the reviewer for pointing out these important references. They have been added to the revised manuscript.

b. I agree that there is a difference in the PVQ defect observed between F1 and F2/F3 generations maintained at 25C, but disagree with the interpretation that penetrance of this defect plateaus at F3 - it appears to reach its maximum effect by F2. Therefore, this result could simply represent a maternal effect (ie: intergenerational) of the parental population being raised at 20C until L4 stage (with gonad development and spermatogenesis occurring at the control temperature).

We agree with the reviewer, as indicated by the title of the paragraph ("Parental conditions have intergenerational effects on axon guidance"). We revised the text to make this point clearer and we changed the plateau timing at F2.

c. I found it difficult to follow the relationship between the effects of heat/maternal age, the heritability of the PVQ defect, and overall H3K4me3 levels. The elevated H3K4me3 levels in animals raised at 25C could just be due to a heat shock response, while the elevated H3K4me3 levels in older adults could be age-related changes in up-regulation of gene expression (as suggested by Pu et al 2018). This interpretation is further supported by the authors' observation that the only other histone modification that changed in aged hermaphrodites was H3K36me3, which is closely associated with active transcription.

We apologize for the lack of clarity in this part of the manuscript. We found intriguing that the defects of PVQs identified in specific environmental conditions (heat, aged mothers but not starvation) were very similar to the one observed in mutants of H3K4me3 regulators (this manuscript and Abay-Nørgaard et al. 2020), and we therefore tested if these conditions impact H3K4me3 levels. It is possible that heat shock response or age-related changes in regulation of

gene expression could potentially explain the elevated H3K4me3 levels at high temperature and in old animals, and we now specify this possibility in the text.

Additionally, although the difference in H3K4me3 levels observed in L4-stage F1 progeny from aged mothers is compelling, but doesn't in itself indicate that the F1 progeny necessarily inherited high levels of H3K4me3. The increased levels of H3K4me could just be an altered response to other defects that come from aged oocytes (like the nutritional differences in these oocytes reported by Perez et al 2017).

We agree with this comment, and we therefore moderate our conclusions in the text.

4. The data presented in Figure 4 do nicely demonstrate that, since the authors did not observe an additive effect in the absence of SET-2 activity for either heat or maternal age, SET-2 is likely responsible for the PVQ defects caused by these environmental exposures.

a. The relationship between SET-2 (and therefore, with H3K4me3 itself) would be strengthened further if they could show that a *set-2* mutation is haploinsufficient only in the presence of heat or advanced maternal age - this experiment would help indicate that removing half of SET-2's activity can sensitive PVQ neurons to environmental effects.

We thank the reviewer for the insightful comments. In figure 4F, *set-2*^{+/-} animals (derived from a WT animal grown at 25 C for two generation crossed with a *set-2* male grown at 20 C) display PVQ defects. Similarly, in figure 4E, *set-2*^{+/-} animals (derived from a 4-days old WT mother crossed with a young *set-2* male) display PVQ defects. Together with the evidence that *set-2*^{+/-} in normal conditions show no PVQ defects (presented in the text,) these results suggest that *set-2* mutation is haplo-insufficient only in the presence of heat or advanced maternal age. We now better clarify this point in the revised manuscript.

b. I found the results shown in Figure 4D-F very interesting, as these data indicate whether the epigenetic effect is in cis (ie: mediated by the histone modification itself) or could be a parental effect in trans (ie: mediated by the M-Z+ or P-Z+ contribution of SET-2 enzyme) - interpreting these results should be added to the discussion, as they may indicate the mechanism of intergenerational inheritance.

We now discuss more extensively these results.

Minor issues:

1. The introduction nicely establishes the research model and summarizes the status of the field. However, to provide a more complete context to understand the major points of this study, it should include more discussion of the following points:

a. It will be important for the authors to fully define inter-generational inheritance versus trans-generational inheritance. This distinction is an important difference in observations presented in this study, and becomes important for interpreting their findings, since the molecular mechanisms underlying each type of inheritance are likely to be distinct.

Agree and done.

b. Page 3 line 15: The phrasing of "grandparental age or tobacco consumption matters in case of neurodevelopmental disorders in grandchildren" is imprecise in a way that obscures the nature of the data. The use of "matters" implies causation. However, because these citations refer to human

epidemiological studies, this should be reworded to accurately reflect the correlative relationship of these studies.

Agreed and done.

c. For the statement on page 4 line 4, the authors should also mention work implicating the importance of H3K4 KDM LSD1/KDM1A in mouse neurodevelopment, particularly because these studies address the potential maternal effect for neurodevelopmental defects in *Kdm1a* M-Z+ individuals (Ancelin et al 2016 and Wasson et al 2016)

Agreed and done.

d. Page 4 line 15: when discussing the role of H3K4me in heritable epigenetic phenotypes, the authors did not include any mention of the first demonstrated instance of transgenerational inheritance of H3K4me (affecting fertility as shown in Katz et al 2009 and Nottke et al 2011) and failed to include discussion of SPR-5's role in longevity, as demonstrated in Greer et al 2016.

Agreed and done.

2. I found the abbreviations "wt des" and "mut des" difficult to read, in part because they look almost like words. Replacing these with "WT-des" and "MUT-des", would help readers parse the text a little more easily.

Agreed and done.

3. Typos:

- a. Page 17 line 3 - "genotyped two identify" should be "to"
- b. P6 line 7 - "descendants carry wild-type alleles" should be "carrying"
- c. P9 line 5 - "guidance hinge on inherited H3K4me3 level" should be "hinges" and "levels"

Agreed and corrected.

Reviewer #2 (Comments to the Authors (Required)):

In figure 1 the authors show that a PVQ axon guidance defect in *set-2* mutants can be inherited. They also show that it is not HRDE-1 dependent. In figure 2 the authors show that there are inherited H3K4me3 and transcriptional changes in descendants of *set-2* mutants. In figure 3, the authors show that temperature and maternal age effect PVQ and other neuron axon guidance, along with corresponding increases in transcription, but starvation does not have an effect. Also, the effects are not inherited in subsequent generations. In figure 4, the authors show that the increased axon guidance effect at elevated temperature and with advanced maternal age is dependent upon *set-2*, but not *set-16*. Finally, the authors show that having a young male or young hermaphrodite in a cross, suppresses the effect of advanced parental age. Also, crossing from a male at normal temperature suppressed the effect of elevated temperature. In both cases, this suppression is dependent upon *set-2*, but not *set-16*.

Overall, this paper has some really interesting observations in a number of different paradigms. It is perhaps somewhat disappointing that some of the most interesting observations are not really followed further (see below). There is also, I believe, a bit of lack of clarity in how the results are presented compared to one another. Specifically, some of the experiments are clearly opposite one

another and should be presented that way. Finally, I think there is a missed opportunity for further conclusion (see below).

Nevertheless, the broad survey presented here of observations related to manipulation of H3K4 methylation are very interesting and will be of broad interest to range of fields. As a result, with some changes to the presentation, I am overall very enthusiastic about this manuscript.

We thank the reviewer for the positive comments.

Overall in figure 1, the effect is somewhat modest. The authors only observe a significant inheritance effect in 7 out of ~40 lines through the hermaphrodite and only 2 when inheritance occurs through the male. Overall, the inheritance through the hermaphrodite at the population level (1D) does look different than the WT (1G), particularly because the overall % of PVQ defects is higher, but the inheritance through the male S1C, does not look different to me than WT (1G). In fact, the average looks to me to be lower upon inheritance through the male (S1C) than WT (1G). The authors need to do population level statistics to determine if there is any difference between 1G vs. S1C, or even 1G vs. 1D, 1E and 1F. For example, it would make some sense if S1C is not different than 1G, that the inheritance can only occur through the oocyte. The authors should also report the average % defect for each, so the comparison can be more directly made. It would also be nice if the authors reported in the text, the exact number of lines with a statistically significant PVQ defect out of the total number of lines assayed.

We thank the reviewer for raising this important point. We now show statistics at population level in Fig. S1G, and present the detailed results in Table S1, as requested. At population level, we found that *set-2*, *ash-2* and *set-2;hrde-1* mutants show significant PVQ defects compared to WT and *unc-6*. We also found that the inheritance through male is not statistically significant at population level, as predicted by the reviewer. In our view, this is not indicative that the inheritance can only occur via the oocyte but rather that there is a requirement to score individual lines when evaluating epigenetic phenotypes, to account for their intrinsic variability and poor penetrance. We now specify this in the revised manuscript.

The control in figure 1 showing that the PVQ defect is not inherited in a mutant with a known PVQ defect that is not epigenetically derived is very nice.

We thank the reviewer for the positive comment

In mutdes of *set-2* mutants the authors observe that the vast majority of lines at F3 no longer have a significant guidance defect, but the defect returns in all lines by F6. This suggests that it requires multiple generations of being homozygous mutant for *set-2* to observe that guidance effect and that it is actually a transgenerational effect itself. The authors sort of suggest this in the discussion, but this interpretation could be clearer. In any case, this is one of the more interesting observations from the manuscript, that is not really followed up. Particularly since a similar observation has been made for the COMPASS complex with regard to longevity (Lee et al eLife 2019). Could the effect here also be due to K9? The authors should definitely discuss this possibility.

We extend our comments on the observation related to the incompletely penetrant phenotype of *set-2* mutant animals at early generations and mention the potential involvement of K9 methylation in discussion.

In wt des, H3K4me3 increased from F3 to F5. This is reminiscent of Greer et al Nature 2011, where lifespan remains extended in descendants of *compass* mutants, despite normal *compass* which

might be expected to restore normal H3K4me3 levels immediately. Could the increase in H3K4me3 levels be necessary for the return to normal lifespan? One way to look at this is to ask whether the increase in H3K4me3 from F3 to F5 is necessary to restore WT H3K4me3. This is certainly suggested by the RNAseq which returns towards WT in F3 and F5 wt des. But is it known that the increase in H3K4me3 in wt des between F3 and F5 returns it more towards normal? The authors probably should have compared H3K4me3 in wt des compared to WT? Perhaps this could be assessed by comparing to published H3K4me3 in L4s? Also, is it possible that the increase in H3K9 observed in Lee et al is what temporarily restricts full H3K4me3 in wt des at F3? The authors could look at this by ChIP or even western blot. On a technical note, in 2C it would be nice to plot (perhaps in the supplement) each F3 vs the corresponding F5 to see if H3K4me3 is always increasing in each individual replicate of the experiment.

We thank the reviewer for this comment, and we agree that comparison with WT L4 samples is beneficial to the manuscript. We have now compared the H3K4me3 ChIPseq with published H3K4me3 data (WT L4) and found that F5 wt des recover more WT L4 peaks than F3 WT-des (Fig. S2E) and show a better correlation to WT L4 samples than F3 WT-des (Fig.S2F). We note that a heatmap of H3K4me3 signal for each F3 and F5 wt des line (Fig. S2A) is already included with the manuscript but lacks specification of lines. We have added it to the figure in the revised manuscript.

We attempted to address this important question experimentally and we were able to obtain set-2 met-2 double mutant. However, double mutant animals exhibit several defects including a high degree sterility, with most of the animals being totally sterile. Double mutant animals show also defects of the vulva, with formation of ectopic vulval invaginations, and a strong lack of coordination, defects that prevent a truthful PVQ analysis. The overall sterility of the double mutant animals prevented us to further exploring the contribution of H3K9 methylation in the propagation of the PVQ phenotype by crossing strategies and the measurement of H3K9me2 levels. In the revised manuscript we mention the possible involvement of H3K9 methylation in the discussion section.

Rogers and Phillips NAR 2020 showed that elevating temperatures opens chromatin, which would be the opposite of mutating set-2. This is consistent with what the authors found in 3E. Therefore, the more interesting question might be to reduce temperature and see if it recapitulates what happens in set-2 mutants?

We thank the reviewer for this interesting suggestion. To fully address this point and study the crosstalk between temperature, chromatin status and epigenetic phenotypes, it will be important to test the PVQ phenotype in parallel to the chromatin landscape by ATAC-seq and ChIP-seq for H3K4 methylation and other histone marks (including H3K9 methylation) at low, standard and high temperatures. We find that this interesting approach deserves a comprehensive analysis which opens up a completely new study.

The authors do show that elevating temperature in set-2 mutants suppresses the effects of elevated temperature, which is consistent with elevated temperature and set-2 mutation acting in opposite directions. But overall there seems to be a lack of clarity with respect to the effect on the axon guidance. At times the authors seem to imply that set-2 mutants, elevated temperature and advanced maternal age are the same because of the similarity in effect on the axon guidance phenotype. But the opposite molecular effects and the suppression clearly indicate that they are different. The main point, it seems to me, is that modulating chromatin transcription in both directions causes the same phenotype. This ultimately probably the most significant conclusion of

the paper, but is somewhat buried. The authors ultimately arrive at this conclusion in the discussion, but the presentation in the results is somewhat confusing.

We apologize for the lack of clarity in this point. We interpreted the results in a slightly different, but not contrasting, manner, suggesting that *set-2* is responsible of the effect on PVQs observed when temperature or maternal age increase and therefore it is acting in the same “pathway” of the conditions. This interpretation is also supported by our RNA sequencing analysis, showing that most deregulated genes at high temperature or in L4 from aged mothers are restored to a normal level in *set-2* mutant animals, suggesting that *set-2* is a downstream effector.

We now mention in discussion the interpretation proposed by the reviewer as a complement to our opinion that higher or lower levels of H3K4me3, compared to wild type, are equally deleterious for the PVQ development and result in aberrant axon migration.

Also, another really important conclusion from this work that is not really emphasized is that, despite both directions affecting axon guidance, there is only epigenetic inheritance of this defect when chromatin and gene expression is decreased in a *set-2* mutant. This is perhaps consistent with the data from Lee et al suggesting that the inherited effect is not due to the lack of H3K4me3, but rather due to the corresponding increase in H3K9 methylation. In my opinion, the paper would be much more impactful if these interpretations are more emphasized in the abstract, discussion and perhaps even the title.

We fully agree with this vision and indeed in our discussion we stated that the effect is transgenerational only when chromatin and gene expression change after *set-2* mutation but not when they are affected by high temperature or aging, conditions that have intergenerational effects on PVQs. This suggests that H3K4me3 changes must be accompanied by other “factors” to result in transgenerational effect. Whether other chromatin changes occur (like K9 methylation, as suggested in Lee et al, 2019), remains, in this context, a speculation waiting for further experimental data. Nevertheless, we now mention this possibility in the revised this possibility.

May 12, 2023

RE: Life Science Alliance Manuscript #LSA-2023-01970-TR

Dr. Anna Elisabetta Elisabetta Salcini
University of Copenhagen
BRIC
Ole Maaloes vej 5
Copenhagen 2200
Denmark

Dear Dr. Salcini,

Thank you for submitting your revised manuscript entitled "Inter- and trans-generational impact of H3K4 methylation in neuronal homeostasis". We would be happy to publish your paper in Life Science Alliance pending final revisions necessary to meet our formatting guidelines.

- please address Reviewer 2's remaining point
- please make sure the author names in your manuscript and our system match
- please add ORCID ID for secondary corresponding author--you should have received instructions on how to do so
- please add a callout for Figure S3D to your text

Figure Check:

- please add sizes next to blots in Figure 3E

A. FINAL FILES:

B. MANUSCRIPT ORGANIZATION AND FORMATTING:

Sincerely,

Reviewer #1 (Comments to the Authors (Required)):

Upon resubmission, this paper is greatly improved - readers now have the evidence needed to evaluate the paper's main claim that H3K4me3 has an epigenetic on *C. elegans* PVQ interneuron axon guidance. The additions and clarifications to the text provide a more tempered approach to discussing their findings, significantly strengthen the paper.

Reviewer #2 (Comments to the Authors (Required)):

In regard to figure 1, the authors did population level statistics on inheritance through males and found that it is not significantly different. I am not sure what the actual P-value was because the authors don't report it (I expect that it may have been nowhere close to significant), but the lack of significance means that there is at least a decent probability that the result is entirely due to chance. In addition, the authors did the experiment ~40 times and only found 2 times where the result was significant. However, instead of making the more straight-forward conclusion that the inheritance is likely occurring only through the oocyte, the authors instead insist that "This result indicates that scoring individual lines when evaluating epigenetic phenotypes is required to account for their intrinsic variability and limited penetrance." This statement is tantamount to saying that statistics don't really apply to epigenetic phenomena. In any case, I don't think this statement is consistent with what the data are saying. So, I would urge the authors to at least state the likely conclusion that inheritance only occurs through the oocyte, even if they want to also want to keep the statement above as an alternate possibility (that is not really supported by the statistics).

Other than that, the changes are satisfactory.

May 17, 2023

RE: Life Science Alliance Manuscript #LSA-2023-01970-TRR

Dr. Anna Elisabetta Salcini
University of Copenhagen
BRIC
Ole Maaloes vej 5
Copenhagen 2200
Denmark

Dear Dr. Salcini,

Thank you for submitting your Research Article entitled "Inter- and trans-generational impact of H3K4 methylation in neuronal homeostasis". It is a pleasure to let you know that your manuscript is now accepted for publication in Life Science Alliance. Congratulations on this interesting work.

DISTRIBUTION OF MATERIALS:

Again, congratulations on a very nice paper. I hope you found the review process to be constructive and are pleased with how the manuscript was handled editorially. We look forward to future exciting submissions from your lab.

Sincerely,
